# Challenges in Integrating Influenza Vaccination Among Older People in National Immunisation Program: A Population-Based, Cross-Sectional Study on Knowledge, Attitudes, Practices, and Acceptance of a Free Annual Program

**DOI:** 10.3390/vaccines13060636

**Published:** 2025-06-12

**Authors:** Mohd Shaiful Azlan Kassim, Rosnah Sutan, Noor Harzana Harrun, Faiz Daud, Noraliza Noordin Merican, Sheleaswani Inche Zainal Abidin, Ho Bee Kiau, Azniza Muhamad Radzi, Nagammai Thiagarajan, Norhaslinda Ishak, Tay Chai Li, Radziah Abdul Rashid, Sally Suriani Ahip, Nor Hazlin Talib, Saidatul Norbaya Buang, Noor Ani Ahmad, Zamberi Sekawi, Tan Maw Pin

**Affiliations:** 1Ministry of Health Malaysia, Putrajaya 62590, Malaysia; shaiful.azlan@moh.gov.my (M.S.A.K.); nharzana@gmail.com (N.H.H.); drnoralizanm@moh.gov.my (N.N.M.); dr.sheleaswani@moh.gov.my (S.I.Z.A.); bkho@hotmail.com (H.B.K.); azniza80@gmail.com (A.M.R.); haslinda84@yahoo.com (N.I.); dr.taychaili@moh.gov.my (T.C.L.); dr.radziah@moh.gov.my (R.A.R.); sally.ahip@gmail.com (S.S.A.); bonafide507@yahoo.com (N.H.T.); s.norbaya@moh.gov.my (S.N.B.); drnoorani@moh.gov.my (N.A.A.); 2Faculty of Medicine, Universiti Kebangsaan Malaysia, Kuala Lumpur 43600, Malaysia; faizdaud@ppukm.ukm.edu.my; 3Malaysian Influenza Working Group (MIWG), Petaling Jaya 40150, Malaysia; zamberi@upm.edu.my (Z.S.); mptan@ummc.edu.my (T.M.P.); 4Faculty of Medicine and Health Sciences, Universiti Putra Malaysia, Serdang 43400, Malaysia; 5Faculty of Medicine, University of Malaya, Kuala Lumpur 50603, Malaysia

**Keywords:** vaccination, Malaysia, influenza, elderly, barriers, knowledge, attitude, practice

## Abstract

**Background:** Influenza poses a significant threat to the health of Malaysians, particularly among the elderly population. It results in high levels of illness and mortality, becoming a financial burden on the government. Vaccination is widely recognised as the most effective measure for controlling the spread and impact of influenza. **Objectives**: This study sought to assess the knowledge, attitudes, and practices (KAP) regarding influenza and influenza vaccination among older adults attending primary healthcare centres in different states of Malaysia. Additionally, the study assessed the level of acceptance for a proposed free annual influenza vaccination program. **Methods**: A nationwide survey was conducted involving 672 older people aged 60 and above who visited nine primary healthcare centres in Malaysia. These centres were selected using proportionate to population size (PPS) sampling to ensure representation from each zone. Participants completed a validated self-reported questionnaire. Descriptive statistics were used to determine the levels of KAP, and a binomial logistic regression model was used to determine the predictors of acceptance for the proposed free annual vaccination program. **Results**: Most participants displayed a strong understanding of influenza illness (74.0%) and the vaccine (65.9%). Moreover, 76.4% of respondents exhibited a positive attitude towards influenza vaccination. However, the prevalence of good vaccination practices was relatively low, with only 29.2% of participants having a history of previous vaccination, and just 55.2% of these consistently practicing annual vaccination. The group acceptance rate for the proposed free annual influenza vaccination was 62.3%. Significant predictors of acceptance included a history of previous vaccination (good practice) (OR = 6.438, 95% CI = 1.16–35.71, *p* < 0.001), a positive attitude towards vaccines (OR = 21.98, 95% CI = 5.44–88.87, *p* = 0.033), and a good level of knowledge about the influenza vaccine (OR = 0.149, 95% CI = 0.03–0.79, *p* = 0.026). **Conclusions**: Increasing the uptake of influenza vaccination among the older population in Malaysia remains a significant challenge. It is recommended that a targeted, free annual influenza vaccination program be implemented for high-risk populations, particularly those with comorbidities and those who have shown greater receptiveness. In addition, health education strategies aimed at raising awareness and understanding of influenza should be prioritised. Strengthening epidemiological data collection and establishing systematic monitoring mechanisms are also essential to support these efforts.

## 1. Background

Influenza infection significantly contributes to annual morbidity, particularly among the elderly. Globally, annual influenza outbreaks are estimated to result in 3 to 5 million cases of severe illness and approximately 290,000 to 650,000 deaths, with elderly individuals being disproportionately affected [1]. The World Health Organization (WHO) recommends that yearly influenza vaccination is the most effective method for preventing the disease, especially among older adults. In 2003, during the World Health Assembly, a target was set to achieve 75% vaccination coverage among older adults by 2010. However, only a few regions have met this goal, prompting an extension of the target. Subsequently, the WHO has shifted its approach from setting specific coverage targets to focusing on broader strategies to improve vaccination coverage and effectiveness [1].

In Malaysia, the influenza vaccination policy strongly advocates for the annual immunisation of individuals aged 60 years and older due to their heightened risk of severe influenza-related complications. The guidelines recommend the most recent updated influenza vaccine as recommended by the World Health Organization (WHO), with a preference for high-dose or adjuvanted formulations to enhance immune response in this vulnerable population [2]. Evidence indicates that influenza vaccination effectively reduces influenza rates in the elderly, with effectiveness ranging from 31% to 58%, based on the types of viruses [3]. Vaccinated individuals in this age group experienced fewer symptoms—between 19% and 45% fewer than their unvaccinated counterparts. Additionally, the vaccine prevented influenza-like illness (ILI) by 13% and reduced all-cause mortality, with influenza vaccine effectiveness (IVE) ranging from 38% to 56% among diabetic patients. In terms of safety, limited evidence suggests that the use of the influenza vaccine shows non-significant adverse effects [3].

Despite substantial evidence supporting the benefits of influenza vaccination, many older people remain hesitant to receive it. Research has identified key factors contributing to this hesitancy, including misinformation, fear of side effects, and a perceived lack of necessity [4,5]. The level of knowledge, attitudes, and practices (KAP) among older adults plays a significant role in their hesitancy and acceptance of influenza vaccination programs. Previous studies highlight that insufficient information and negative attitudes are major barriers to vaccination [6]. Understanding these KAP factors is crucial for assessing the readiness of the program strategies to improve vaccination rates and reduce the impact of influenza. Therefore, this study aims to evaluate the KAP related to influenza (ILI) and influenza vaccination among elderly individuals visiting primary healthcare centres in Malaysia and to explore how these factors influence their willingness to accept the proposed free annual influenza vaccination program.

## 2. Materials and Methods

### 2.1. Study Design

This study employed a cross-sectional, multistage sampling design conducted across nine primary healthcare centres in Malaysia. These centres were strategically selected to represent each geographical zone in Malaysia (northern, central, southern, eastern, and Borneo), as well as varying states and localities (urban, semi-urban, and rural), as detailed in Table 1. The sample size was calculated using a standard formula for comparing two proportions, with the largest proportion of 50% used to suggest 383 participants were needed [7]. However, the sample size was adjusted to 670 to account for the design effect and potential non-responses. The sample size required for each primary healthcare centre was determined using proportionate to population size (PPS) sampling, with participants recruited through a quota sampling technique.

### 2.2. Data Collection

The study population comprised older people visiting the centres for treatment, accompanying family members, or other purposes. Data were collected through face-to-face interviews using structured questionnaires administered by trained field staff. Inclusion criteria required participants to be 60 years or older, Malaysian, fluent in either Malay or English, have good cognitive function, and be capable of reading.

The questionnaire used in the study was validated in earlier research with Cronbach’s alpha >0.70, showing good internal reliability among visitors of primary healthcare centres in Saudi Arabia and diabetic patient in tertiary hospital in South Africa [8,9]. To ensure accuracy, the instrument was translated into Malay and subsequently back-translated to English to assess any discrepancy in understanding the meaning for the items asked. The content was evaluated and assessed by a team of experts, comprising family health physicians, epidemiologists, and statisticians. The Malay version was tested with 30 participants in one of the selected primary healthcare centres to confirm it was clear and understandable, and it was found to have good internal reliability for the sections on knowledge, attitudes, and practices. The final version of the questionnaire was structured into four main sections: sociodemographic (5 questions), knowledge (11 questions), attitudes (7 questions), and practices (4 questions). Each section contained questions with multiple-choice answers.

### 2.3. Data Analysis

To analyse the results, the same cut-off scores were used as in previous studies [8,9]. A score of 65% or above in influenza and vaccination knowledge was considered ‘good’, whereas scores below 65% were classified as ‘poor’. The attitude variable was categorised as ‘positive’ if there were four or more positive responses or ‘negative’ if there were four or more negative responses. For practices, a history of having received an influenza vaccine was categorised as ‘good’ practice. Acceptance of the proposed free annual vaccination program was assessed with the question: “If the Ministry of Health Malaysia provides you with free influenza vaccination every year, would you take it?” The responses were ‘Yes’, ‘Maybe’, or ‘No’.

Data entry and statistical analysis were carried out using IBM Corp.’s Statistical Package for the Social Sciences (SPSS) version 23, with a significance level set at ≤0.05. Descriptive statistics were employed to summarise categorical variables using frequencies and percentages and continuous variables using means and standard deviations. A binomial logistic regression model was used to determine the predictors of acceptance for the proposed free annual vaccination program. The model considered responses of ‘Yes’ as the dependent variable, with ‘No’ and ‘Maybe’ as the reference category. The binomial logistic regression estimates how much more (or less) likely participants are to say “Yes” instead of “No” or “Maybe” when certain predictor variables are present. The independent variables included sociodemographic factors such as gender, age, ethnicity, marital status, education level, and comorbidities, as well as knowledge, attitudes, and practices related to influenza and vaccination.

### 2.4. Ethical Consideration

Ethical approval for the study was granted (ID: NMRR ID-23-03546-MYP). Informed consent was secured from all participants, with assurances of confidentiality, anonymity, and voluntary participation. All personal details were handled with strict confidentiality.

## 3. Results

Our study enrolled 672 participants with an average age of 68.4 ± 7.2 years. The majority were female (64.3%), Malay (63.5%), married (65.0%), had secondary and tertiary level education (64.3%), and had comorbidity (73.4%) (see Table 2).

### 3.1. Knowledge Regarding Influenza and Influenza Vaccine

The findings showed that most participants demonstrated a solid understanding of influenza (74.0%) and the influenza vaccine (65.9%). Figure 1 shows the distribution of correct answers for each question, which revealed that the majority believed that influenza can spread from one to another (90.9%), can be prevented (90.1%), and was caused by a virus (89.5%). Interestingly, only 52.5% claimed that influenza is not a common cold, and 62.1% of participants believed that influenza occurs throughout the year in Malaysia (Table 2). The symptoms reported were running nose (96.3%), fever (90.0%), cough (89.7%), and sore throat (85.6%).

In our sample, 89.7% of respondents indicated that the vaccine could prevent influenza, and 89.3% believed it was safe. Additionally, 83.4% were aware of a vaccine that prevents influenza, with most participants (98.6%) noting that it is administered via injection. Interestingly, only 58.2% of participants acknowledged that the vaccine might have side effects. The majority (68.0%) believed that the vaccine provides protection for a year, and 84.5% thought it could prevent influenza-related complications. Nearly all participants (96.4%) agreed that the vaccine should be taken before the onset of influenza (Figure 2).

### 3.2. Attitude Towards Influenza Vaccine

Regarding participants’ attitudes toward the influenza vaccine, 76.4% displayed a positive attitude. The majority of participants (76.0%) agreed serious complications can be prevented by the vaccine, and 75.1% expressed willingness to receive the vaccine if it effectively prevents influenza. Interestingly, while 66.9% acknowledged the importance of annual vaccination, 60.7% simultaneously believed they personally did not need the vaccine due to natural immunity, highlighting a common discrepancy between public health awareness and personal risk perception (Figure 3).

### 3.3. Practice Towards Influenza Vaccine

There were 29.2% of participants who had a good practice (having ever received an influenza vaccine). Among them, only 55.2% received annual vaccination, 8.2% every two years, 3.3% every three years, and 33.3% of them only received once in a lifetime (Table 2). Among those who did not receive vaccination previously, the main reasons were the vaccine was expensive (46.35%) and that it had serious side effects (33.9%). The data also estimated 7.2% of participants had history of admission for influenza infection. Finally, the prevalence of acceptance of the free annual influenza vaccination was 62.3% (Figure 4).

### 3.4. Predictors of Acceptance to the Free Annual Influenza Vaccination

A binomial logistic regression model was employed to estimate the prediction model through a two-step process. In the first step, the basic model was estimated with acceptance of the free annual influenza vaccination program as the dependent variable and sociodemographic variables (age, gender, ethnicity, education level, marital status, and comorbidity) as covariates. In the second step, the main predictors—knowledge about influenza and vaccines, attitude, and practice—were added to the model. The final model specification was statistically significant (x^2^: 52.778; *p* < 0.001), with Pseudo R^2^ (Nagelkerke) = 0.559. The final model identified significant predictors, including history of previous vaccination (having ever received the influenza vaccine before) (OR = 6.438, 95% CI = 1.16–35.71, *p* < 0.001), positive attitude towards vaccines (OR = 21.98, 95% CI = 5.44–88.87, *p* = 0.033) and good knowledge on influenza vaccine (OR = 0.149, 95% CI = 0.03–0.79, *p* = 0.026) (see Table 3).

## 4. Discussion

This study found that among older people in Malaysia, there was a high level of good knowledge and positive attitude towards influenza vaccines. However, less than one-third had a history of influenza vaccination, and only half of those individuals received the vaccine annually. Notably, 62.3% of older persons expressed acceptance of the free annual vaccination program. Factors such as a history of previous vaccination and a positive attitude significantly predicted acceptance.

In tropical regions like Malaysia, influenza can occur year-round without clear seasonal patterns, leading to unpredictable outbreaks. Influenza A (including subtypes like H1N1 and H3N2) is more prevalent than influenza B, although the prevalence of both types varies significantly from year to year. This trend aligns with global patterns. Influenza A viruses mutate rapidly, resulting in frequent outbreaks and higher transmission rates compared to less variable influenza B viruses [10]. The high mutation rate of influenza A necessitates annual vaccine updates to match current strains. Currently, Malaysia has approximately 3.7 million older people (aged 60 and above) [11]. Recent studies suggest declining vaccine uptake among vulnerable populations post-COVID-19 pandemic [12]. Given these trends, is Malaysia adequately prepared for a free annual influenza vaccination program?

Epidemiological data are crucial for monitoring and evaluating vaccination programs. They help track disease incidence, assess vaccine effectiveness, and identify areas needing intervention. However, Malaysia faces challenges in accurately estimating influenza incidence due to inconsistent sentinel surveillance implementation across regions and diagnostic limitations. Not all patients with influenza-like illness (ILI) undergo testing, and reliance on clinical diagnosis can lead to inaccurate case estimates. Enhanced surveillance systems and improved diagnostics are essential [13].

Research on the KAP is vital. In the current study, only 68.0% of participants knew the influenza vaccine provides one-year protection. Targeted educational campaigns are needed to improve understanding and encourage timely vaccination among the elderly. Despite high knowledge and positive attitudes, actual vaccination rates remain low. Only 29.2% had a history of previous influenza vaccination, and adherence to annual schedules was suboptimal. A neighbouring country, such as Singapore, reported a higher prevalence of 32.4% [14], while other countries documented even higher rates, such as 77.8% in South Korea [15], 94.1% in Mexico, and 51.0% in Japan [16].

This study has also identified perceived cost as the primary barrier to vaccination. Despite this, only 62.3% of the population expressed willingness to accept the proposed free annual influenza vaccination. Interestingly, other studies reported a high acceptance rate among healthcare workers, where approximately 74% indicated willingness to receive the vaccine among twelve low- and middle-income countries [17]. However, the current vaccination coverage among Malaysian healthcare workers stands at only 26.3% [18]. This disparity underscores the significant challenges in achieving high vaccination coverage in the general population.

Finally, the current study revealed that the significant predictors of acceptance of the free annual influenza vaccination program were a history of previous vaccination and a positive attitude towards vaccines. This is consistent with previous research [4,6]. Additionally, a review of the literature indicates that insufficient information and negative attitudes significantly reduce the likelihood of influenza vaccination. Conversely, sufficient knowledge about influenza and a positive attitude towards vaccination are strongly associated with higher vaccination rates [8].

### 4.1. Recommendation

Based on the current findings, it is recommended that a targeted annual influenza vaccination program be implemented specifically for high-risk populations, particularly elderly individuals with comorbidities such as cardiovascular disease, diabetes, and chronic respiratory diseases. These individuals are considered to be at a significantly higher risk of severe complications from influenza. Notably, it has been observed that those with comorbid conditions have demonstrated greater compliance with vaccination programs, making them an ideal focus for such targeted initiatives [19]. The program should be implemented opportunistically for individuals who are receptive and willing to participate voluntarily, as well as for all older adults preparing for pilgrimage.

It is recommended that an influenza vaccination program be implemented for the pre-elderly working population (ages 50–59) at panel clinics, with the costs fully covered by their respective companies rather than the government. Having the vaccine covered under occupational health coverage through panel clinics is expected to significantly increase vaccination rates by reducing barriers such as cost and accessibility. Recent studies have highlighted the effectiveness of workplace vaccination programs in enhancing employee health and reducing the economic burden associated with flu-related illness [20]. Moreover, it is suggested that instilling the practice of receiving the influenza vaccine at a pre-elderly age can encourage continued vaccination into older age, fostering long-term health benefits and reducing the burden on healthcare systems [21].

While the previous suggestions address immediate needs, it is also crucial to focus on long-term strategies for enhancing influenza vaccination uptake among all older population. Government policies should address current barriers such as accessibility, affordability, and awareness to improve vaccination rates. To overcome these barriers, it is recommended that strategies such as offering the vaccine for free in all government facilities and providing a subsidised rate for private facilities be implemented to alleviate financial burdens and encourage higher vaccination uptake. Financial support should be directed towards reducing the cost of vaccines and ensuring that they are available at no cost to the older population.

Evidence supports that multifaceted approaches, combining financial incentives with educational efforts, are effective in increasing vaccination rates among the older population [18]. Therefore, it is recommended that KAP and awareness programs be strengthened and targeted. The population survey reported a high prevalence of comorbidities and a low level of health literacy among the rural elderly population in Malaysia [19]. Information, education, and communication (IEC) are considered crucial in enhancing influenza vaccine acceptance among the older persons. Effective IEC strategies can address common concerns, misconceptions, and gaps in knowledge that may impede vaccine uptake. Training programs for healthcare workers should be incorporated to ensure they are well informed and competent to provide accurate information, address concerns effectively, and motivate older people to participate in vaccination programs. It is also recommended that the influenza vaccination program be integrated into the current program, i.e., scheduled vaccination during regular healthcare visits or through community health programs. Effective communication such as reminder systems, including automated phone calls, text messages, and mail reminders, have been demonstrated to significantly improve vaccination coverage and reduce influenza-related complications among the elderly [20,21].

Finally, public health strategies focused on empowerment, engagement, and collaboration are considered crucial in enhancing influenza vaccine acceptance among older people [21]. Targeted communication efforts that engage community leaders and utilise multiple channels such as social media, print materials, and face-to-face interactions can help reach a broader audience, build trust, and overcome barriers to vaccination. Collaboration between healthcare providers, community organisations, and public health authorities enhances the effectiveness of vaccination efforts by pooling resources, sharing expertise, and creating a unified approach to addressing barriers. The effective integration of these social and healthcare domains is essential for addressing the complex needs of older adults, particularly in Asian contexts where family and community care play significant roles [22]. This collective effort not only improves vaccine acceptance rates but also strengthens overall public health resilience.

### 4.2. Strength and Limitations

One of the key strengths of this study is that it is the first to comprehensively assess the knowledge, attitudes, and practices (KAP) related to influenza vaccination, along with the acceptance of a proposed free annual influenza vaccination program among older persons in Malaysia. By conducting a large-scale national survey across multiple states and using a robust sampling method, the study provides a representative and in-depth understanding of the factors influencing vaccine uptake in this high-risk population. To our knowledge, this is the first nationwide, population-based study in Malaysia to assess the KAP and acceptance of a proposed free influenza vaccination program among the elderly. It offers new insights into how prior vaccination history, health literacy, and attitude interrelate in predicting vaccine uptake, which can inform targeted public health interventions. Unlike previous studies, this work also suggests a novel phased approach targeting the pre-elderly and high-risk groups as a sustainable national immunisation strategy.

However, the study had certain limitations. A significant one was the reliance on self-reported data, which could be influenced by recall bias or social desirability bias, possibly impacting the accuracy of the responses, i.e., the inclusion criteria for good cognitive function was informally assessed by the participant’s ability to comprehend and complete the questionnaire. However, as no formal cognitive screening tool was used; this represents a potential limitation. This study does not explore in detail the barriers to acceptance of influenza vaccination, which would require a mixed-method approach, including qualitative research. The study only captured data from nine primary healthcare facilities within Malaysia. As a result, the study was geographically restricted, which may limit the ability to apply the findings to broader populations. Additionally, rural representation was limited to a single health clinic, which may underrepresent the diversity of rural populations and affect the generalisability of findings to those settings. Furthermore, the study was limited to individuals (patients, family members, and visitors) who accessed the facilities during the study period, which leaves the work with challenges in interpreting the results for generalisation. Moreover, since the majority of participants were younger elderly individuals, female, and had comorbidities, caution should be taken when applying these findings to the broader Malaysian elderly population. This could have introduced bias into the results, potentially skewing them in a certain direction. Additionally, the cross-sectional design of the study prevents any conclusions about causality.

## 5. Conclusions

It was found that most participants possessed good knowledge about influenza and its vaccine and demonstrated positive attitudes towards vaccination. However, despite these favourable attitudes, only 29.2% had a history of influenza vaccination, with only half of those individuals having received the vaccine annually. It was reported that 62.3% of older persons expressed acceptance of the free annual vaccination program. The prediction model identified that a history of previous vaccination and a positive attitude were significant predictors for the acceptance of the free annual influenza vaccination.

Based on these findings, it is recommended that a targeted annual influenza vaccination program be implemented, aimed at high-risk populations, particularly those with comorbidities and individuals who are already more receptive to vaccination. To improve vaccine uptake, it is essential that comprehensive awareness and health education strategies be employed while the current surveillance system is strengthened.

In addition, the findings underscore the need for continued research and programmatic innovation. Future efforts should explore the use of digital tools, community-based outreach, and integration with existing public health programs to enhance vaccination uptake among underrepresented groups, particularly those with lower educational attainment or residing in rural areas. Qualitative studies are also warranted to better understand personal beliefs, misconceptions, and cultural influences that may hinder vaccine acceptance. A multifaceted, inclusive approach that combines behavioral insight, policy support, and healthcare system integration will be key to achieving sustainable improvements in influenza vaccination coverage among the Malaysian elderly.

## Figures and Tables

**Figure 1 vaccines-13-00636-f001:**
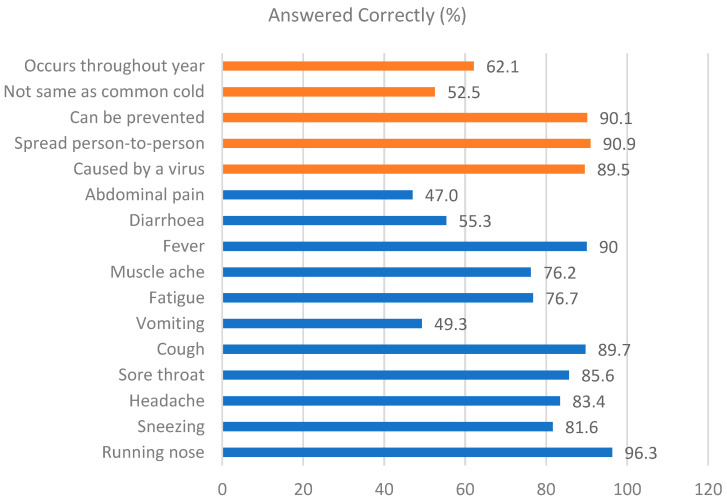
Participants’ knowledge regarding influenza (n = 672).

**Figure 2 vaccines-13-00636-f002:**
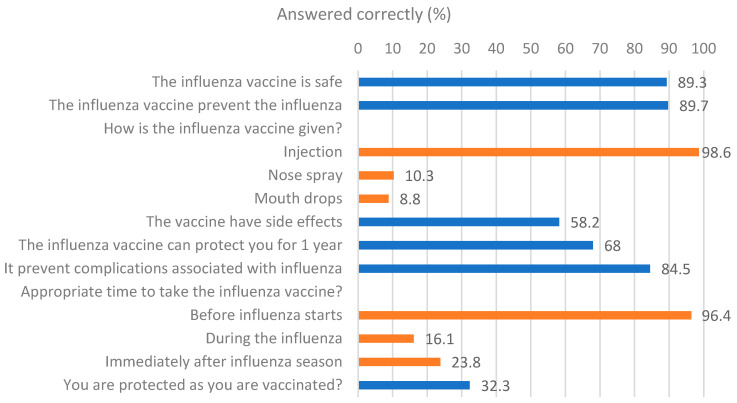
Participants’ knowledge regarding influenza vaccine (n = 672).

**Figure 3 vaccines-13-00636-f003:**
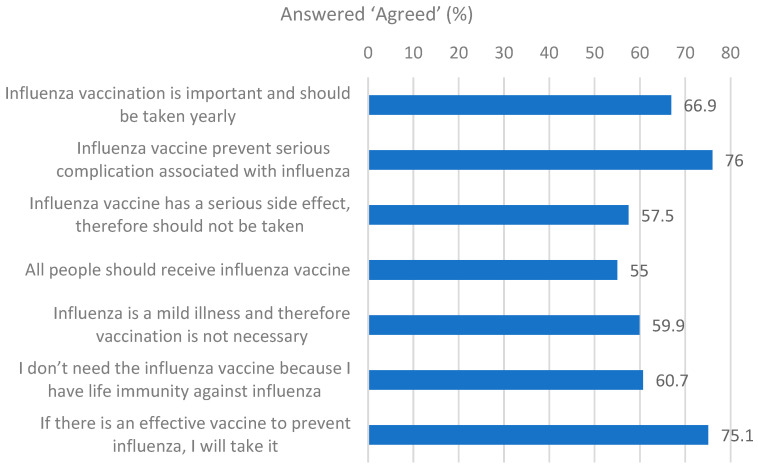
Participants’ attitudes regarding influenza vaccination (n = 672).

**Figure 4 vaccines-13-00636-f004:**
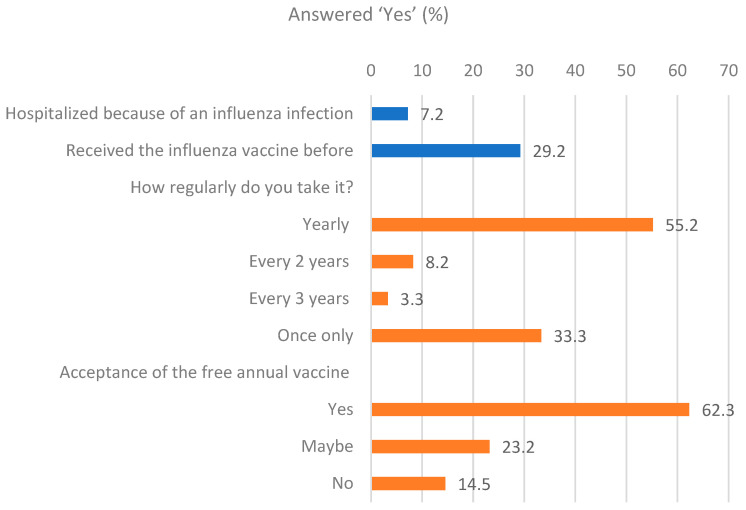
Participants’ practice towards influenza vaccination (n = 672).

**Table 1 vaccines-13-00636-t001:** Primary health centres selected by zone.

Zone	State	Primary Health Centre	Locality
East	Pahang	Karak Health Clinic	Semi-Urban
Central	Selangor	Bandar Botanic Health Clinic	Urban
		Pandamaran Health Clinic	Semi-Urban
	WP Kuala Lumpur	Kuala Lumpur Health Clinic	Urban
		Batu 9 Hulu Langat Health Clinic	Semi-Urban
North	Kedah	Guar Chempedak Health Clinic	Rural
	Perak	Simpang Health Clinic	Semi-Urban
South	Melaka	Ayer Keroh Health Clinic	Urban
Borneo	Sarawak	Kota Samarahan Health Clinic	Urban

**Table 2 vaccines-13-00636-t002:** Sociodemographic characteristics of participants, N = 672.

Characteristics	n	Mean (SD)/(%)
**Age**		68.43 (7.16) min 60, max 100
**Sex**		
Male	240	35.7
Female	432	64.3
**Ethnicity**		
Malay	427	63.5
Chinese	151	22.5
Indian	67	9.9
Bumiputera Sarawak and Sabah	25	3.7
Others	3	0.4
**Marital status**		
Married	437	65.0
Widow/widower	190	28.3
Separated/divorced	29	4.3
Single	16	2.4
**Education level**		
No formal education	53	7.9
Primary	187	27.8
Secondary	286	42.6
Tertiary	146	21.7
**Comorbidity**		
Present	493	73.4
Absent	179	26.6

**Table 3 vaccines-13-00636-t003:** Predictors of acceptance of the free annual influenza vaccination program.

Acceptance ‘Yes’	*p* Value	Odd Ratio	95% Confidence Interval for Odds Ratio	
			Lower Bound	Upper Bound
Good knowledge on influenza	0.790	1.265	0.224	7.140
Poor knowledge on influenza				
Good knowledge on vaccine	**0.026**	**0.149**	0.028	0.792
Poor knowledge on vaccine				
Positive attitude	**<0.001**	**21.982**	5.438	88.869
Negative attitude				
Good practice	**0.033**	**6.438**	1.161	35.712
Poor practice				

The reference category includes ‘No’ and ‘Maybe’. x^2^: 60.912; *p* < 0.001, Pseudo R^2^ (Nagelkerke) = 0.559, −2 Log Likelihood = 63.923. Variable(s) entered in Step 1: age, gender, ethnicity, education level, marital status, and comorbidity.

## Data Availability

The data that support the findings of this study are available from the corresponding author upon reasonable request.

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
