# Peer review of "Challenges in Integrating Influenza Vaccination Among Older People in National Immunisation Program: A Population-Based, Cross-Sectional Study on Knowledge, Attitudes, Practices, and Acceptance of a Free Annual Program"

_vaccines, 2025, doi:10.3390/vaccines13060636_

Round 1
Reviewer 1 Report
Comments and Suggestions for Authors
Please see the attachment.

Please see the attachment.
Author Response
We would like to thank the reviewer for the constructive feedback. We have revised the manuscript accordingly and addressed all comments as detailed below:
Comment 1: In page 1, line 23, “becoming financial burden on the government” should be "becoming a financial burden on the government".
Response: Thank you for pointing this out. We have corrected the sentence in the Abstract to read: “...and becoming a financial burden on the government.” please refer to page 1, line 23,
Comment 2: In page 2, line 60, Should “recommended” be in the present simple tense?
Response: The sentence originally read: “The World Health Organization (WHO) recommended that yearly influenza vaccination is the most effective method…”
We have revised it for clarity and tense consistency: “The World Health Organization (WHO) recommends that yearly influenza vaccination is the most effective method…”
Comment 3: In page 3, line 103, Where does the data come from?
Response: Thank you for your observation. We have clarified the source of data collection in Section 2.2 (Data Collection), adding: “The study population comprised older people visiting the centres for treatment, accompanying family members, or other purposes. Data were collected through face-to-face interviews using structured questionnaires administered by trained field staff.” Please refer to page 3, line 106 to 107.
Comment 4: What are the innovative points of this paper? Compared with published papers, what are the core differences in terms of their manifestations?
Response: We appreciate the request for clarification on the novelty of this work. We have added the following statement at the end of the "Strength and Limitations" section:
"To our knowledge, this is the first nationwide, population-based study in Malaysia to assess KAP, and acceptance of a proposed free influenza vaccination program among the elderly. It offers new insights into how prior vaccination history, health literacy, and attitude interrelate in predicting vaccine uptake which can inform targeted public health interventions. Unlike previous studies, this work also suggests a novel phased approach targeting the pre-elderly and high-risk groups as a sustainable national immunization strategy." Please refer to page 10 - 11, line 326 - 332.
Additional Suggestion: Carefully check the grammar problems in the article again.
Response: We have thoroughly revised the manuscript for grammar, clarity, and consistency. Changes include:
-
Correction of article usage and verb tenses.
-
Alignment of subject-verb agreements.
-
Consistency in terminologies and phrasing.
Reviewer 2 Report
Comments and Suggestions for Authors
The main aim of this article was to evaluate the practices related to influenza (ILI) and influenza vaccination among elderly individuals and to explore how these factors influence their willingness to accept the proposed free annual influenza vaccination program. We have to emphasize, that the main aim was achieved.
But we recommend for authors to reconsider the approach to evaluation of results.
For example: subsection 3.2 - affirmation "66.9% acknowledged the importance of getting the vaccine annually, while 60.7% believed they did not need the vaccine due to their natural immunity against influenza" seems contradictory. It looks like that the same people have contradictory opinions. Perhaps, one of these affirmations should be eliminated.
Another point: perhaps the authors have to exclude 7,9% respondents without any formal education. That might to improve the overall results of this investigation.
Author Response
Reviewer Comment:
The main aim of this article was to evaluate the practices related to influenza (ILI) and influenza vaccination among elderly individuals and to explore how these factors influence their willingness to accept the proposed free annual influenza vaccination program. We have to emphasize, that the main aim was achieved. But we recommend for authors to reconsider the approach to evaluation of results. For example: subsection 3.2 - affirmation "66.9% acknowledged the importance of getting the vaccine annually, while 60.7% believed they did not need the vaccine due to their natural immunity against influenza" seems contradictory. It looks like that the same people have contradictory opinions. Perhaps, one of these affirmations should be eliminated.
Response:
We thank the reviewer for the positive acknowledgment that the main aim of the study was achieved. We appreciate the concern regarding the seemingly contradictory results in subsection 3.2.
The statements in question refer to two different attitude items that were independently assessed in the survey. The first item measured awareness of general public health importance of annual vaccination, while the second reflected personal beliefs regarding susceptibility or immunity. It is indeed possible and has been noted in behavioral science literature that individuals may acknowledge the public importance of a health intervention while simultaneously perceiving themselves as personally exempt. This cognitive dissonance is common in health behavior research and highlights the complexity of vaccine hesitancy. To avoid confusion, we have revised the sentence in subsection 3.2 to provide a clearer explanation: “Interestingly, while 66.9% acknowledged the importance of annual vaccination, 60.7% simultaneously believed they personally did not need the vaccine due to natural immunity highlighting a common discrepancy between public health awareness and personal risk perception.” Please refer to page 6, line 180 - 183.
Reviewer Comment:
Another point: perhaps the authors have to exclude 7.9% respondents without any formal education. That might improve the overall results of this investigation.
Response:
We appreciate this suggestion. However, we respectfully maintain the inclusion of respondents without formal education in our analysis for the following reasons:
-
Representativeness: The goal of our study was to reflect the real-world diversity of Malaysia’s elderly population. Excluding a specific subgroup, such as those without formal education, would reduce the representativeness and generalizability of our findings.
-
Public Health Relevance: This subgroup may be especially vulnerable due to lower health literacy and may require tailored health communication strategies. Including them provides valuable insights into disparities and can inform equitable vaccine promotion strategies.
That said, we have clarified in the manuscript that subgroup analyses by education level were performed, and that this variable was included as a covariate in our regression analysis to control for its potential influence on outcomes.
Reviewer 3 Report
Comments and Suggestions for Authors
Thank you for your submission! I have a couple of suggestions for your consideration:
- Just a few minor grammer corrections needed (Page 1 line 23 add the word "a" financial burden.." page 7 line 184 change to "that had a good practice" currently reads "had good a practice"
- Were the particpants given a list of possible co-morbitites so they understood what that term meant?
- How was good cognitive function measured? Or was it self-reported that they that had good cognitive function ( a limitation that you did point out about self-reporting may not be totally accurate)
- A possible limitation is that you only had one rural clinic but had 4 each of urban and semi-urban clinics- you mentioned geographic areas a possible limitation but I think also just having one rural may also be a limitation
- Consider stating "the most recent updated influenza vaccine recommended by WHO" rather than stating quadravalent or trivalent (page 2 line 68) to avoid confusion. The WHO has recommended only the trivalent for 24-25 and 25-26 seasons.
- Table 2 consider having separated/divorced rather than separate & divorced
- Table 3 P-value are typically not reported as zero but <0.001
- References are all good but some websites were accessed a year ago- WHO infuenza recommendations differ from 23-24 season to current recommendations
It is a very important research topic and thank you for this!
Author Response
Reviewer Comment:
Just a few minor grammar corrections needed (Page 1 line 23: add the word “a financial burden”; page 7 line 184: change to “that had a good practice” currently reads “had good a practice”).
Response:
Thank you for your attention to detail. We have made the necessary grammatical corrections as follows:
-
Page 1, line 23: Revised to “becoming a financial burden on the government.”
-
Page 7, line 187: Corrected to “had a good practice.”
Reviewer Comment:
Were the participants given a list of possible comorbidities so they understood what that term meant?
Response:
Yes, participants were provided with a list of commonly known chronic conditions such as hypertension, diabetes, heart disease, asthma, and stroke, in both lay and medical terms. This was done to ensure that they understood what constitutes a comorbidity.
Reviewer Comment:
How was good cognitive function measured? Or was it self-reported that they had good cognitive function (a limitation that you did point out about self-reporting may not be totally accurate).
Response:
We appreciate this important observation. Cognitive function was assessed informally through the ability of participants to understand and complete the questionnaire independently or with minimal assistance. It was not measured using a standardized cognitive screening tool and was thus a subjective criterion. We have now clarified this in the Limitation section: “Good cognitive function was informally assessed by the participant’s ability to comprehend and complete the questionnaire. However, as no formal cognitive screening tool was used, this represents a potential limitation.” Please refer to page 11, line 335 - 338.
Reviewer Comment:
A possible limitation is that you only had one rural clinic but had four each of urban and semi-urban clinics you mentioned geographic areas as a possible limitation, but I think also just having one rural may also be a limitation.
Response:
Thank you for this helpful suggestion. We agree and have amended the Limitations section to specifically include this:
“Additionally, rural representation was limited to a single health clinic, which may underrepresent the diversity of rural populations and affect the generalizability of findings to those settings.” Please refer to page 11, line 342 - 344.
Reviewer Comment:
Consider stating “the most recent updated influenza vaccine recommended by WHO” rather than stating quadrivalent or trivalent (page 2, line 68) to avoid confusion. The WHO has recommended only the trivalent for 2024–2025 and 2025–2026 seasons.
Response:
We have revised the sentence accordingly to reflect the latest WHO recommendation and reduce confusion:
“The guidelines recommend the most recent updated influenza vaccine as recommended by the World Health Organization (WHO), with a preference for high-dose or adjuvanted formulations to enhance immune response in this vulnerable population.” Please refer to page 2, line 68 - 70.
Reviewer Comment:
Table 2: consider using ‘separated/divorced’ instead of ‘separated & divorced’.
Response:
This suggestion has been implemented. The category now reads “Separated/Divorced” for consistency and clarity.
Reviewer Comment:
Table 3: p-values are typically not reported as zero but as <0.001.
Response:
Thank you. We have revised all such values in Table 3 to “<0.001” in accordance with standard reporting practices.
Reviewer Comment:
References are all good, but some websites were accessed a year ago — WHO influenza recommendations differ from 2023–2024 season to current recommendations.
Response:
We appreciate this remark. We have updated the relevant WHO references to reflect the most recent 2024–2025 and 2025–2026 influenza vaccine composition recommendations. The access dates have also been updated to June 2025. please refer to Reference section, page 12, line 397.
Reviewer Comment:
It is a very important research topic and thank you for this!
Response:
We sincerely thank you for the kind words and constructive feedback. We believe the suggested improvements have further strengthened the clarity, accuracy, and impact of our manuscript.